# Preliminary Study: DNA Transfer and Persistence on Non-Porous Surfaces Submerged in Spring Water

**DOI:** 10.3390/genes14051045

**Published:** 2023-05-06

**Authors:** Morgan L. Korzik, Josep De Alcaraz-Fossoul, Michael S. Adamowicz, David San Pietro

**Affiliations:** 1Department of Forensic Science, University of New Haven, West Haven, CT 06516, USA; 2College of Agricultural Sciences and Natural Resources, University of Nebraska, Lincoln, NE 68583, USA

**Keywords:** DNA analysis, DNA transfer and persistence, trace DNA, latent fingermarks, DNA quantification, STR detection, short tandem repeat detection, forensic genetics, forensic science

## Abstract

Submerged items are often thought to lack evidentiary value. However, previous studies have shown the ability to recover DNA from submerged porous items for upwards of six weeks. The crevices or interweaving fibers in porous items are thought to protect DNA from being washed away. It is hypothesized that, because non-porous surfaces do not have the same traits that might aid in DNA retention, then DNA quantities and the number of donor alleles recovered would decrease over longer submersion periods. Additionally, it is hypothesized that DNA quantity and the number of alleles would be negatively affected by flow conditions. Neat saliva of known DNA quantity was applied to glass slides and exposed to stagnant and flowing spring water to observe the effects on both DNA quantity and STR detection. Results supported that DNA deposited onto glass and subsequently submerged in water experienced a decrease in DNA quantity over time, yet submersion did not have as strong of a negative effect on the detected amplification product. Additionally, an increase in DNA quantity and detected amplification product from designated blank slides (no initial DNA added) could indicate the possibility of DNA transfer.

## 1. Introduction

Submerged items can occur in a variety of forensic situations from underwater to indoor crime scenes. At indoor scenes, investigators may find items left behind in sinks, showers, toilets, and bathtubs. Even if a crime occurs elsewhere, a perpetrator may try to destroy evidence by throwing it into a river, lake, or ocean [1]. Many investigators believe that submerged items no longer hold evidentiary value, especially for identification purposes [2]. However, identifiable trace evidence, fingerprints, and DNA have all been successfully recovered from submerged items. After the introduction of RFLP (Restriction Fragment Length Polymorphism), DNA has become one of the most prominent forensic techniques [3], but research suggests that its recovery potential from aquatic conditions is dependent on numerous factors [4].

In addition to the physical presence of water, DNA recovery on submerged items seems to be further hindered by higher salinities and stronger currents [4]. The structure of the items appears to influence DNA recovery as well. Porous items, such as bedsheets and paper, appear to retain submerged DNA for longer times than non-porous items, such as car doors and bottles [4]. The crevices of fibers existing in porous items may be able to protect DNA from being lost to currents [4]. Cellular material surrounding DNA may be able to protect DNA for a given time, but the water itself is capable of hydrolyzing DNA and destroying valuable evidence [5,6].

However, submerged non-porous items can still hold evidentiary value after this restrictive timeframe. Multiple studies have shown that latent fingerprints can be recovered from non-porous items [7,8,9] even after a month of submersion [10]. Similar to DNA, submerged latent fingerprint recovery appears to be further hindered by higher temperatures, higher salinities, and faster currents [10,11,12]. When depositing a latent fingerprint (also referred to as a fingermark), one leaves behind natural oils and insoluble secretions which may be less miscible in water than soluble DNA. Fingerprint samples are capable of generating low levels of DNA, although the nature of that DNA (shed keratinocytes, epithelial cells, nucleated cells from other parts of the body that the hands have contacted, and cell-free DNA as examples) is still debated [13,14]. Some refer to this recovery of low-level DNA as “touch DNA”, or “transfer DNA” yet the term “trace DNA” will be utilized in this study.

Because fingerprints are retained on submerged non-porous evidence, it was hypothesized that the insoluble components of fingerprint secretions may protect DNA by trapping any residual epithelial cells between the secretions and the surface itself, permitting trace DNA recovery even under submerged conditions. Previous studies have highlighted the ability to obtain DNA from latent fingerprints [15,16,17,18]. Investigators and analysts could utilize this knowledge to determine the cost–benefit of pursuing friction ridge comparison in addition to, or possibly in lieu of, trace DNA analysis. Such knowledge could supplement research that focuses on choosing optimal samples for trace DNA analysis. For example, nucleic acid dyes, such as SYBR Green I, have been utilized to visualize low-level DNA on touched objects to evaluate the cost–benefit of pursuing trace DNA analysis [19].

The behavior of DNA in saliva deposition slide samples was evaluated in addition to the mentioned fingerprint samples. Trace samples have an inherent unpredictability in the amount of DNA that will transfer in deposition, so saliva slide samples provided a way of being able to deposit a known amount of DNA and monitor loss over time. Therefore, saliva deposition slide samples served as controlled samples, while fingerprint slide samples were representative of mock casework samples. Because non-porous surfaces do not have traits that might aid in DNA retention, it was hypothesized that DNA quantities and the number of donor alleles recovered would decrease over longer submersion periods. Additionally, it was hypothesized that DNA quantity and the number of alleles would be negatively affected by flow conditions.

## 2. Materials and Methods

### 2.1. Experimental Phases

The experiment was broken into four phases. In Phases 1 and 2, slides containing the saliva samples were placed into stagnant and flowing spring water, respectively. In Phases 3 and 4, slides containing the fingerprint samples were placed into stagnant and flowing spring water, respectively. Phases 1 and 2 pertain to controlled samples with known initial quantities of DNA, while Phases 3 and 4 reflect mock casework samples where the initial quantity of DNA is unknown. Additionally, two replicates of blank slide samples per submersion time were utilized for each phase.

### 2.2. Water Environments

Two water environments were utilized: stagnant spring water and flowing spring water. All water environments were created in an indoor laboratory setting with Poland Spring^®^ water, as the company provides information about the levels of various chemical and biological components and physical characteristics of the water. Before experimentation, samples from the Poland Spring^®^ water jugs were extracted and quantified for the possible presence of background human DNA, and none was detected for the methods utilized (see Section 2.5 below).

The stagnant environment was created by filling a glass tray with approximately 1.5 L spring water. The glass tray was cleaned prior to use by spraying the tray with 20% bleach, 70% reagent alcohol, and distilled water followed by cross-linking for 1000 s with a SpectroLinker XL-1500 UV Crosslinker. While in use, the glass tray was stored under a PCR hood (Figure 1).

The flowing environment was created by filling a horizontal flume with approximately 5 L spring water. The flume was formed with 2-inch diameter polyvinyl chloride (PVC) plastic, with overall dimensions measuring 32 by 56 inches at its widest points. The flume was powered by a PULACO aquarium wave-making pump with a maximal flow rate of 50 GPH (Gallons per Hour). A few inches after the pump were deemed a “dead zone” where no samples were placed, permitting the flow to rise to a constant speed of approximately 10 cm/s. PVC was cut horizontally in the “sample zone” to allow samples to be placed into and removed from the water. Samples were oriented such that the wide surface of the slides was parallel to the direction of the current for the samples to experience a constant flow. The water recirculated through the flume over the course of the experiment. To avoid disrupting the flow when rounding the corners, the flume utilized two 45-degree PVC elbows rather than 90-degree angles. Between uses, the flume was cleaned by spraying with 20% bleach, 70% reagent alcohol, and distilled water. Due to its size, the flume was unable to be cross-linked and was stored in an active research room when in use. Bins were placed over open areas in the flume to attempt to avoid extraneous contamination (Figure 1).

### 2.3. Creating and Depositing Samples

Sirchie glass microscope slides were utilized as the non-porous surface. The glass slides were cross-linked on both sides for 1000 s in a SpectroLinker XL-1500 UV Crosslinker before use. Three random slides were swabbed and processed for human DNA after cross-linking. No quantity of human DNA was recovered, so it was assumed that cross-linking for 1000 s on both sides would remove any background human DNA from the slides.

Sample slides were divided into three main categories. “Blank” slide samples had nothing added to the slide after cross-linking. For Phases 1 and 2, “deposition” slide samples had 5 μL fresh neat saliva added to the slide. A portion of the neat saliva (20 μL) was extracted following the QIAamp DNA Investigator Protocol: Isolation of Total DNA from Small Volumes of Blood or Saliva and quantified (yielding a concentration of approximately 6 ng/μL) before deposition to estimate the initial amount of DNA being deposited onto each sample. Two blank slide samples and five deposition samples were utilized per submersion time per water condition for these phases. For Phases 3 and 4, “fingerprint” slide samples had a sebaceous fingerprint deposited onto the slide with a force of approximately 400 g with contact lasting three seconds. Force was measured by placing the slide on a scale when depositing the fingerprint. The right thumb of one donor was utilized as the fingerprint source for all samples. Before depositing the fingerprints, the donor washed their hands for 20 s with warm soapy water and allowed their hands to air-dry for approximately 5 min. The donor then wiped their right thumb across their forehead to obtain sebaceous secretions and deposited the print onto the designated glass slide. The cleaning process was repeated between fingerprint depositions. Two blank slide samples and five fingerprint slide samples were utilized per submersion time per water condition for these phases. Due to the inherent unpredictability of touch DNA samples, the initial amount of DNA transferred from the finger to the slide was unknown. However, three fingerprint samples (right thumb) were created, extracted, quantified, and amplified for human DNA prior to submersion to attempt to establish a baseline. The quantities of these initial fingerprint samples ranged from 0.0106 to 0.0343 ng/μL with an average of 0.0224 ng/μL (σ^2^ = 1.404 × 10^−4^).

One male donor aged 22–24 was utilized in this study and was the donor for both saliva and fingerprint samples. All samples were obtained according to informed consent.

### 2.4. Sample Submersion and Recovery

For Phases 1 and 2, all blank and deposition slide samples entered the water at the same time, and a set of slides were removed at each time interval: 6 h, 12 h, 24 h, 48 h, 72 h, and 168 h (1 week). For Phases 3 and 4, all blank and fingerprint slide samples entered the water at the same time, and a set of slides were removed at each time interval: 24 h, 48 h, 72 h, and 168 h (1 week). An additional set was evaluated at 0 h to serve as an initial assessment. The experiment was carried out in an indoor lab environment at room temperature (approximately 20 °C).

In the flowing environment, cleaned blank slides were inserted in place of the removed slides to avoid altering the way water flowed through the flume. To reduce the risk of introducing microbes when removing the sets, researchers utilized proper PPE and disinfected gloves with 70% reagent alcohol.

As slides were removed from the water, they were placed on a sterile benchtop to air dry. When fully dry, the slides were either swabbed and extracted for DNA or stored at −80 °C until extraction. When stored at −80 °C, slides were kept in cross-linked microscope slide holder boxes. The wet-dry swab method was utilized to collect potential DNA from the slides which is a common technique for trace DNA recovery [19,20].

### 2.5. DNA Analysis

Qiagen’s QIAamp DNA Investigator Kit was utilized to extract the DNA collected from each sample. All swabs from samples (blank slides, deposition slides, and fingerprint slides) were extracted following the QIAamp DNA Investigator Protocol: Isolation of Total DNA from Surface and Buccal Swabs. This extraction procedure included adding proteinase K and Buffer ATL to the sample, incubating at 56 °C with shaking for two hours, adding Buffer AL, incubating at 70 °C with shaking for ten minutes, adding ethanol, and transferring the lysate to a QIAamp MinElute column. Samples were washed with Buffer AW1, Buffer AW2, and ethanol before eluting from the QIAamp MinElute column utilizing 50 μL Buffer ATE. Used glass slides and swab heads were saved and stored at −80 °C in case needed for further analysis. Samples were stored in 1.5 mL Eppendorf tubes at −80 °C after extraction.

The Quantifiler^TM^ Human DNA Quantification Kit with a QuantStudio 5 Real-Time PCR System (Design and Analysis Software v1.5.1) was utilized to quantify the human DNA present in samples. The Quantifiler^TM^ Human DNA Quantification Kit has a lower threshold of detection of approximately 16 pg/μL of human genomic DNA [21]. The quantity of human DNA present on all samples was recorded.

STR amplification was performed for both submersion conditions at every submersion time on all blank slide samples, three saliva slide deposition, and three fingerprint slide samples. Applied Biosystems’s GlobalFiler^TM^ PCR Amplification Kit was utilized to amplify twenty-four STR loci. Capillary electrophoresis was performed with ILS600 LIZ Size Standard on an Applied Biosystems 3130xl Genetic Analyzer (16 capillary array 50 cm, POP4 polymer, 1× buffer). The injection conditions utilized were 3 kV/5 s, and the run conditions were 15 kV/1500 s. STR amplification results were interpreted with the GeneMarker HID STR Human Identity Software (version 2.4). Sample amplification data recorded included number and designation of alleles present, individual allele peak heights, and heterozygous peak height ratios.

Internal controls were utilized and during all steps (extraction, quantification, and amplification) in order to monitor any contamination or error. Negative internal controls yielded no indication of contamination.

### 2.6. Statistical Methods

*T*-Tests (two-tailed) were utilized to calculate *p*-values as estimates of statistical significance (*p* < 0.05 in all instances) when comparing one condition to another (for example, comparing DNA quantities recovered at 24 h in the flowing versus stagnant water conditions). Other mathematical analyses included calculating averages and variance values (σ^2^) or standard deviation values based on the samples for both DNA quantity and the number of alleles recovered.

## 3. Results

### 3.1. Phases 1 and 2: DNA Quantity

In general, DNA quantities of deposition slide samples in both stagnant and flowing conditions decreased over the one-week submersion period (Figure 2). Initial deposition slide samples were created by adding neat saliva to cleaned glass slides, letting the slides air-dry, and swabbing with the wet-dry swab method. Initial DNA quantities from the slides ranged from 2.9653 to 5.1249 ng/μL with an average of 3.96142 ng/μL (σ^2^ = 0.8360).

When comparing the DNA quantities from deposition slide samples obtained between the stagnant versus flowing conditions for each submersion time, there was little statistical significance (*p* = 0.824, *p* = 0.810, *p* = 0.016, *p* = 0.332, *p* = 0.20, and *p* = 0.028 for each submersion interval, respectively, significance level *p* < 0.05). However, at 24 h and at 1 week, DNA quantities were statistically significantly higher in the stagnant condition than in the flowing (*p* = 0.016 and *p* = 0.028, respectively). These results may indicate that DNA behaves similarly in stagnant and flowing conditions at shorter submersion times, but with prolonged submersion, flow can negatively affect DNA recovery. When comparing the DNA quantities from deposition slide samples obtained for each submersion time versus the initial DNA quantities, DNA quantities were statistically significantly lower in both stagnant and flowing conditions for every time period (Stagnant: *p* = 0.0073, *p* = 0.0033, *p* = 0.0006, *p* = 4.3 × 10^−5^, *p* = 2.1 × 10^−5^, and *p* = 1.3 × 10^−5^; Flowing: *p* = 0.0006, *p* = 0.0009, *p* = 0.006, *p* = 0.0001, *p* = 6.7 × 10^−5^, and *p* = 4.7 × 10^−5^; for each submersion interval, respectively, significance level *p* < 0.05). Although longer submersion times negatively affected DNA recovery, this suggests that a portion of DNA is lost from samples even with shorter submersion times.

In general, DNA quantities of blank slide samples in both stagnant and flowing conditions increased over the one-week submersion period (Figure 3). A subset of three blank slides were swabbed after cross linking and before submersion. The initial DNA quantities from these blank slides were undetermined with Quantifiler^TM^ Human, so the DNA quantities were assumed to be zero.

Of note, multiple blank slide samples were of sufficient quantity that the samples were expected to pass the amplification threshold. For instance, at one week, the highest DNA quantity recovered from a blank slide sample was approximately 0.07 ng/μL, which could theoretically provide 1.05 ng DNA for amplification, therefore, meeting the target of approximately 1 ng. Statistical methods were not applied to blank samples due to the limited sample size to make any determinations of statistical significance. However, the recovery of sufficient DNA quantities on multiple (initially blank) samples over various submersion times is noteworthy, particularly in the implications for forensic DNA analysis which will be discussed further.

### 3.2. Phases 1 and 2: Number of Donor Alleles Recovered

A reference sample from the donor was utilized to gather information about the donor’s alleles at AMEL and 17 STR loci. In total, 30 alleles of the donor were considered.

For deposition slide samples, it was noted when one of the 30 donor alleles had dropped out from the electropherogram. For blank slide samples, it was noted when one of the 30 donor alleles had dropped into the electropherogram.

Any foreign alleles detected were also noted and were not included in the calculation of the number of donor alleles recovered. When foreign alleles were detected, alleles consistent with the donor were also detected, and foreign alleles produced less signal in the electropherograms. There were no samples where alleles were detected without any being consistent with the donor. Foreign alleles were not common but were most prevalent in the flowing water condition which may have been due to the assembly of the horizontal flume, the inability to cross-link the horizontal flume due to its size, the storage of the horizontal flume in a shared laboratory space, or other additional factors. For the sake of the study, it was assumed that foreign alleles were not shared with donor alleles.

In the stagnant condition, on average, over 70% of the donor alleles were recovered from the deposition slide samples at every submersion interval over one week (Figure 4). In the flowing condition, on average, over 70% of the donor alleles were recovered at every submersion interval except at one week (Figure 4). There were no statistically significant differences when comparing the number of donor alleles recovered in stagnant versus flowing conditions for all submersion intervals (*p* = 0.579, *p* = 0.292, *p* = 0.169, *p* = 0.329, *p* = 0.588, and *p* = 0.252 for each submersion interval, respectively, significance level *p* < 0.05).

In the stagnant condition, donor alleles were not detected on the blank slide samples until 24 h submersion. No donor alleles were detected on the subset of initial blank slide samples prior to submersion. The number of donor alleles recovered varied over the course of one week, but at one week submersion, on average, over 50% of the donor alleles were recovered from the blank slide samples in stagnant water (Figure 4). In the flowing condition, donor alleles were detected on the blank slide samples starting at 6 h submersion. However, the average number of donor alleles recovered was below 25% for each submersion interval in flowing water (Figure 4). Large standard deviations in the number of donor alleles recovered from blank slide samples may reflect the unpredictability in DNA transfer and persistence (Figure 4).

When comparing the number of donor alleles recovered on deposition slide samples at all submersion times to the initial 30 alleles, the number of alleles detected was statistically significantly lower than the initial at 72 h and one week submersion in stagnant (*p* = 0.024 and *p* = 0.02, respectively, significance level *p* < 0.05). In flowing water, the number of donor alleles recovered at 12 h was statistically significantly lower than the initial (*p* = 0.038, significance level *p* < 0.05), but there was no statistically significant difference at any other submersion interval. This may indicate that enough DNA to yield a sufficient STR profile may be retained on non-porous surfaces even after multiple days submersion.

### 3.3. Phases 3 and 4: DNA Quantity

In general, DNA quantities of fingerprint slide samples in both stagnant and flowing conditions decreased over the one-week submersion period (Figure 5). Initial DNA quantities from the fingerprint slide samples ranged from 0.0106 to 0.0343 ng/μL with an average of 0.0224 ng/μL (σ^2^ = 1.404 × 10^−4^). These initial quantities were lower than that of the deposition samples associated with Phases 1 and 2, so the performance of fingerprint slide samples was not compared to that of the deposition slide samples.

When comparing the DNA quantities from fingerprint slide samples obtained between the stagnant and flowing conditions for each submersion interval, there was no statistical significance (Stagnant: *p* = 0.639, *p* = 0.485, *p* = 0.153, and *p* = 0.297; Flowing: *p* = UND, *p* = 0.423, *p* = UND, and *p* = 0.423; for each submersion interval, respectively, significance level *p* < 0.05). When comparing the DNA quantities from fingerprint slide samples obtained for each submersion interval versus the initial fingerprint DNA quantities, there was not a significant decline in DNA quantity until the 72 h submersion duration (Stagnant: *p* = 0.57, *p* = 0.64, *p* = 0.9, and 0.02; Flowing: *p* = 0.98, *p* = 0.17, *p* = 0.04, and *p* = 0.006; for each submersion interval, respectively, significance level *p* < 0.05). For the flowing water, fingerprint DNA quantities were statistically significantly lower at 72 h and 1 week than the initial values (*p* = 0.04 and *p* = 0.006). For the stagnant water, fingerprint DNA quantities were statistically significantly lower at 1 week than the initial values (*p* = 0.02). This indicates that fingerprint samples may experience loss of DNA at shorter submersion times in flowing water in comparison to stagnant, supporting that flow negatively affects DNA recovery.

In general, DNA quantities of blank slide samples in both the stagnant and flowing conditions were similar over time. All quantities for blank slide samples were below 0.006 ng/μL and, therefore, were expected to fall below the target amplification threshold of 1 ng.

### 3.4. Phases 3 and 4: Number of Alleles Recovered

The average number of donor alleles recovered from fingerprint slide samples differed for all submersion intervals in the stagnant condition (Figure 6). The average percent of donor alleles recovered was 62.22% at 24 h (σ^2^ = 2670.37), 47.78% at 48 h (σ^2^ = 1048.15), 81.11% at 72 h (σ^2^ = 114.81), and 10.00% at 1 week (σ^2^ = 44.44). Variance values were highest at the first submersion time, and less variation was observed with longer submersion. High variation within the earlier submersion times may be reflective of the unpredictability of touch and low-quantity samples prior to any influence from the stagnant water. In the flowing condition, no donor alleles were recovered from fingerprint slide samples. Despite this, visible ridge detail was visible in all fingerprint samples at each submersion time in either condition (Figure 7).

No donor or foreign alleles were recovered from the blank slide samples for any of the submersion times in either water condition.

Although the donor had 30 known alleles, three fingerprint slide samples were deposited and collected for DNA analysis without any submersion in water. The three samples contained 73.3%, 20%, and 20% of the 30 known donor alleles, respectively. For the purposes of Phase 3 and 4, the initial average number of alleles was considered to be 37.8% (11.34 alleles) of the 30 donor alleles. When comparing the number of donor alleles recovered on fingerprint slide samples at all submersion times to the initial average number of alleles (37.8% of the 30 donor alleles), the number of alleles detected was not statistically significantly different at any submersion time for either condition (Stagnant: *p* = 0.52, *p* = 0.718, *p* = 0.246, and *p* = 0.201 for each submersion interval, respectively; Flowing: *p* = 0.168 at all submersion intervals; significance level *p* < 0.05). However, the lack of statistically significant differences may be partially attributed to the inherent inconsistency in depositing a full DNA profile through fingerprint deposition.

## 4. Discussion

### 4.1. Revisiting Hypotheses

For Phases 1 and 2, overall DNA quantity of deposition slide samples decreased over one week submersion. However, minimal allele drop-out was observed for deposition slide samples, and some instances of allele drop-in were observed for the blank slide samples. Although longer submersion times resulted in a decrease of DNA quantity, one week submersion did not significantly affect the ability to recover donor alleles from deposition slide samples.

For Phases 3 and 4, overall DNA quantity of fingerprint slide samples decreased, with the decline being statistically significant only at 72 h and one week submersion (*p* = 0.04 and *p* = 0.006, respectively, significance level *p* < 0.05). Multiple instances of allele drop-out were observed for fingerprint slide samples. Over one week submersion, fingerprint slide samples in flowing water experienced a decline in the number of alleles recovered while fingerprint slide samples in stagnant water did not.

For Phases 1 and 2, overall DNA quantities were lower in flowing than in stagnant water, but the two submersion intervals where the difference was statistically significant were at 24 h and one week submersion (*p* = 0.016 and *p* = 0.028, respectively, significance level *p* < 0.05). There was no difference in the number of donor alleles recovered between stagnant and flowing conditions. Flow did affect DNA quantities at some submersion times but did not hinder the ability to recover donor alleles from deposition samples.

For Phases 3 and 4, there was no difference in the DNA quantities between stagnant and flowing conditions. The presence of fingerprint secretions possibly aided in retaining DNA to the slides even in flow conditions. The number of donor alleles recovered from fingerprint slide samples were lower in flowing than in stagnant water for all submersion times. Although flow did not affect DNA quantities of fingerprint samples, flow hindered the ability to recover donor alleles.

### 4.2. Limitations

For this study, there were five replicates of deposition or fingerprint slide samples and two replicates of blank slide samples per submersion time for DNA quantitation purposes. Additionally, there were three replicates of deposition or fingerprint slide samples and two replicates of blank slide samples per submersion time for STR amplification purposes. Ideally, the sample size would have been increased for both DNA quantitation and STR amplification. Only one donor was utilized in this study, and the same individual donated DNA and fingerprints. Incorporating more individuals could account for variability among donors, especially with the fingerprint samples. Both limitations are due to time and budget constraints.

### 4.3. Implications

#### 4.3.1. Trace DNA from Fingerprint Samples

One of the main goals of the study was to evaluate the performance of fingerprint samples in terms of the retention of both fingerprint secretions and DNA. The quantity of DNA transferred from the donor’s finger to the glass surfaces was lower than anticipated at all submersion times, including the initial time when the fingerprints were not submerged. The highest quantity was collected at 48 h submersion (0.0055 ng/μL in stagnant water), yet this value was not significantly different from any of the other DNA quantities collected from fingerprints for any of the submersion times. Despite difficulty recovering DNA after one week of submersion, fingerprints recovered from both stagnant and flowing water contained visible ridge detail without the need for enhancement throughout the duration of the experiment. These findings support that of previous studies where fingerprints were able to be recovered from submerged non-porous surfaces up to 42 days of submersion, with the highest success being within a week of submersion [10]. The results of this particular study indicate that if a fingerprint is observable on a non-porous item recovered from the water, then investigators should consider prioritizing friction ridge comparison over DNA recovery, especially if recovered from flowing environments.

#### 4.3.2. Possibility of Transfer

In designing the experiment, the goal of incorporating blank slide samples to be removed at each submersion time was to act as a control measure for detecting background DNA in the water throughout the duration of the experiment. Increasing DNA quantities of blank slide samples, particularly in Phases 1 and 2, indicated that some degree of drop in was occurring in the blank slide samples, as no human DNA was detected on blank slides prior to submersion. However, STR detection revealed alleles consistent with the donor from the deposition slide samples on the blank slides. Although foreign alleles were detected on some of the blank slide samples in the flowing condition, alleles consistent with the donor were recovered in addition to those foreign ones. Because negative internal controls did not yield any detectable DNA in either quantification or STR detection, it was assumed that any alleles present on the blank slide samples were a result of transfer rather than contamination. During the experiment, donor DNA from the deposition slide samples may have been transferred into the water before transferring onto the blank slide samples.

Transfer refers to the mode by which DNA arrives on an item. DNA can arrive on an item through direct contacts, such as when saliva was directly applied to deposition slides in Phases 1 and 2 (Figure 8). Indirect transfer can occur when DNA deposited on Surface A arrives on Surface B, such as DNA from deposition slide samples arriving on blank slide samples (Figure 8). DNA may also undergo multiple degrees of transfer, such as secondary to tertiary to quaternary and so on [22]. DNA transferring from deposition slide samples to the water to the blank slide samples would be considered a tertiary transfer as DNA is transferring from Surface A (deposition slide) to B (water) to C (blank slide) (Figure 9).

For Phases 1 and 2, transfer onto blank slide samples in stagnant water appeared to be most prominent at one week. Without knowledge of the sample identification, the electropherograms of deposition and blank slide samples at one week would be extremely difficult to distinguish (Figure 10 and Figure 11). With how the experiment was designed, all deposition and blank slides were submerged into the same water vessel at the same moment, and then after designated submersion intervals, sets of slides were removed. Initially, it was anticipated that any DNA lost from the deposition samples to the surrounding water would be hydrolyzed and suffer in quality. Additionally, transfer from the deposition slide samples to the blank slides was not anticipated. In theory, DNA lost from deposition slide samples yet not destroyed in the water could have cross-transferred to the other deposition slide samples or transferred onto the blank slide samples. Slides in the set removed at one week could have been subjected to higher DNA concentrations in the water than the other sets. In stagnant water, the higher DNA concentrations may have eventually settled onto the blank slides, resulting in high yields of donor alleles. In flowing water, transfer onto blank slide samples was most prominent at 6 h and then relatively consistent for the remainder of the submersion times. Because samples in the flowing condition were experiencing a constant flow, any DNA that was lost to the water and transferred onto blank slide samples may have been pushed off again and continued to circulate in the horizontal flume.

Transfer onto blank slide samples was not observed in Phases 3 and 4. This may be due to the fingerprint slide samples containing a lesser amount of DNA prior to submersion than in Phases 1 and 2. Additionally, components in saliva may have aided in DNA persistence in Phases 1 and 2. Another possibility is that if DNA were lost from the fingerprint samples in Phases 3 and 4, the DNA may be repelled from any remaining oils in fingerprint secretions, preventing lost DNA from redepositing and cross-transferring onto other fingerprint samples in the same water condition.

Due to multiple samples being submerged in the same water together, the transfer observed is not necessarily reflective of a situation where DNA is indirectly transferred from Surface A to Surface B without cross-transfer from additional surfaces containing DNA from the same donor. DNA transfer observed in the flowing water was under recirculating conditions (similar to pools, tanks, and fountains), and free-flowing conditions (similar to rivers and streams) may have different possibilities for DNA transfer. For instance, DNA lost from Surface A in a free-flowing condition may be swept away before having the opportunity to transfer to Surface B. Under recirculating conditions, DNA lost from Surface A may have the opportunity to deposit back onto Surface A, transfer to Surface B, and remain in the water as examples. Additionally, DNA transfer through water may be affected by the volume of water, direction of current, and consistency of flow rate, in addition to other environmental factors, including salinity, pH, contaminants, UV light, physical obstacles in the environment, etc. However, the ability to recover donor DNA from the blank slide samples at least indicates the ability of DNA to transfer and persist under submerged conditions without providing a statement of probability. While perhaps not to the extent of what was observed in this study, these results indicate that investigators and analysts should at least be wary of overstating the significance of recovering submerged DNA. For instance, the increased sensitivity of forensic DNA analysis to detect low levels of DNA has, in turn, permitted the recovery of DNA not deposited by direct contact [22]. By assuming the recovery of DNA from an item, even if a full profile, was a result of the individual directly contacting the item, an analyst can overstate the significance of detecting that DNA.

### 4.4. Future Possibilities

Future studies could incorporate metal and plastic items of various smooth and textured surfaces to determine if any surface combinations provide the best conditions for DNA recovery. Additionally, future studies could evaluate how submersion in water in addition to other environmental factors influences DNA recovery which could provide a well-rounded view to submerged DNA behavior in natural environments. Introducing soap or detergents as variables may also be informative for submerged items recovered at indoor crime scenes. The results of an additional study could aid investigators in determining if a non-porous item recovered from water should be considered for DNA analysis, taking into account the item’s surface and environmental conditions.

## 5. Conclusions

In general, for controlled samples, DNA deposited onto glass and subsequently submerged in water experienced a decrease in DNA quantity over time, yet submersion did not have as strong of a negative effect on the amplification product. On average, over half the donor alleles were recovered at every submersion interval for both the stagnant and flowing conditions. The average percentage of donor alleles recovered was above 70% until the one-week submersion interval. In some instances, particularly at earlier submersion intervals, all donor alleles were recovered from the deposition slide samples. In most instances, flowing conditions either further negatively influenced DNA recovery or had little influence, in comparison to stagnant water alone. Of note is that in the mock casework samples, flow did not hinder DNA quantity, but the introduction of flow resulted in no alleles being recovered from the fingerprints despite alleles being recovered in the stagnant condition. Results of the controlled samples indicate that the evidentiary value of submerged non-porous items should not be discounted. Although DNA quantities and the number of alleles recovered were low throughout the mock casework samples, there were still instances where fingerprints yielded valuable DNA after submersion. For example, the average number of donor alleles recovered from the fingerprints after 72 h submersion was over 80% in the stagnant water condition. Even when trace DNA was not recovered from the mock casework samples, friction ridge detail was visible for all fingerprints without enhancement, even after one week submersion. This supports that friction ridge comparison may be more informative than or beneficial to perform in addition to trace DNA analysis for these sample types. Overall, trace DNA does appear to be recoverable from non-porous items after submersion; however, prolonged submersion can result in evidence loss.

Blank slide samples experienced an increase in DNA quantity and the number of donor alleles over time. Because the blank slide samples were initially cleaned and a subset was swabbed, extracted, and quantified with no detectable DNA, the blank slides were assumed to be free of background human DNA. Due to this, in addition to all internal negative controls showing no contamination, the presence of these alleles was thought to be due to DNA transfer. Specifically, the possibility of transfer from deposition slide samples to the water to the blank slide samples which could be defined as an example of tertiary transfer. Investigating DNA transfer was not an initial objective for this study and was instead an observation when evaluating results. Due to this, statistical analyses could not be conducted on the limited number of blank slide samples. However, the observations of this study indicate the need to explore DNA transfer through water further. In multiple instances, deposition and blank slide samples yielded visually similar electropherograms, some to the extent that one would be unable to distinguish the deposition and blank slide samples without existing knowledge of which sample was which. In forensic DNA analysis, one is unable to know with certainty if trace DNA recovered from an evidence item was a result of direct contact or transfer, highlighting the importance of understanding the possibility of DNA transfer for interpretation purposes.

The accumulation of results indicate that the behavior of low-level DNA and conditions for indirect DNA transfer may be unpredictable. Further, interpretations concerning the significance of DNA recovery from items submerged in water should not be overstated.

## Figures and Tables

**Figure 1 genes-14-01045-f001:**
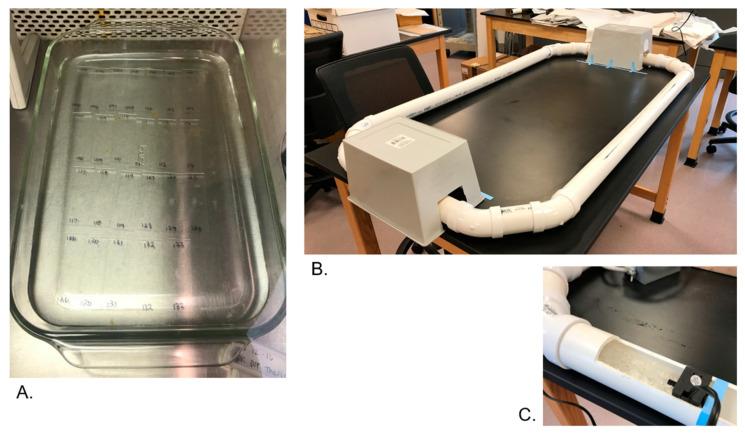
Water environments created in lab, where (**A**) shows slides submerged in the stagnant water condition under a PCR hood; (**B**) shows the flowing water condition created from PVC plastic in a shared lab space; and (**C**) shows a closer view of the PULACO aquarium wave-making pump that powered the current in the flowing water condition.

**Figure 2 genes-14-01045-f002:**
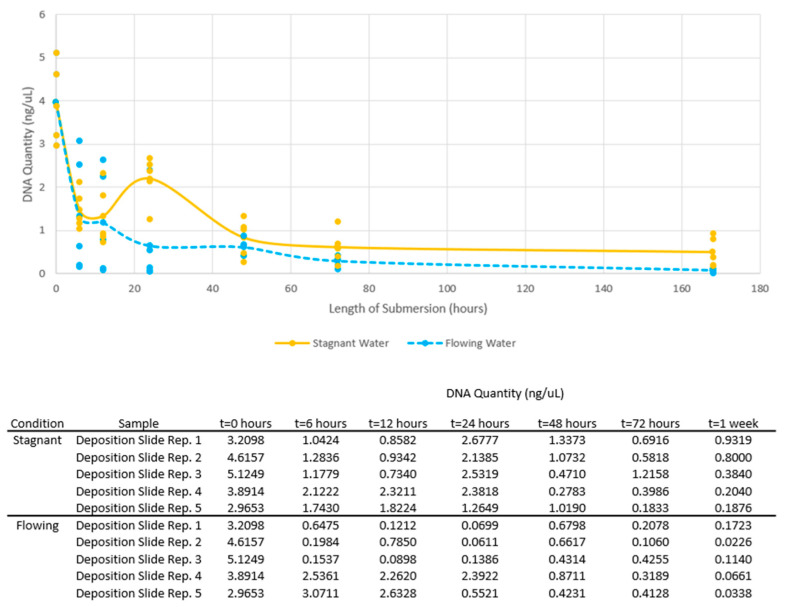
DNA quantities of deposition slide samples in stagnant and flowing water over one week submersion. Dots represent individual quantity values for all samples, while lines connect the average quantity values per submersion time. The table shows the DNA quantity obtained for each replicate sample.

**Figure 3 genes-14-01045-f003:**
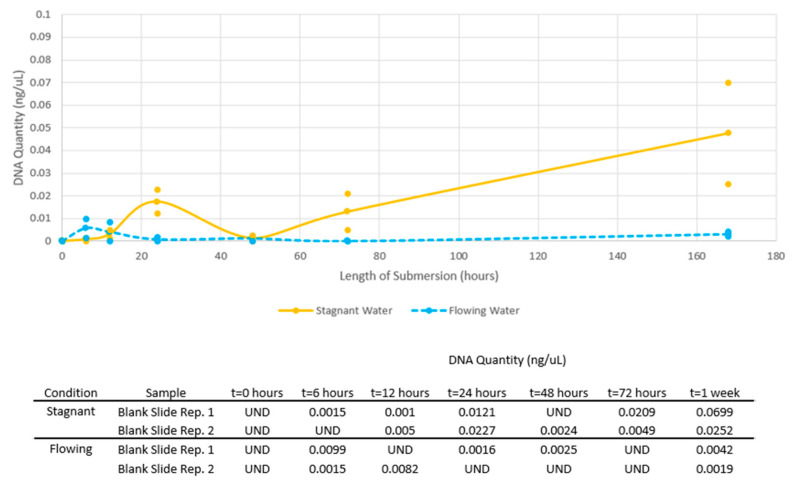
DNA quantities of blank slide samples in stagnant and flowing water over one week submersion. Dots represent individual quantity values for all samples, while lines connect the average quantity values per submersion time. The table shows the DNA quantity obtained for each replicate sample. “UND” refers to an undetected value reported at quantification.

**Figure 4 genes-14-01045-f004:**
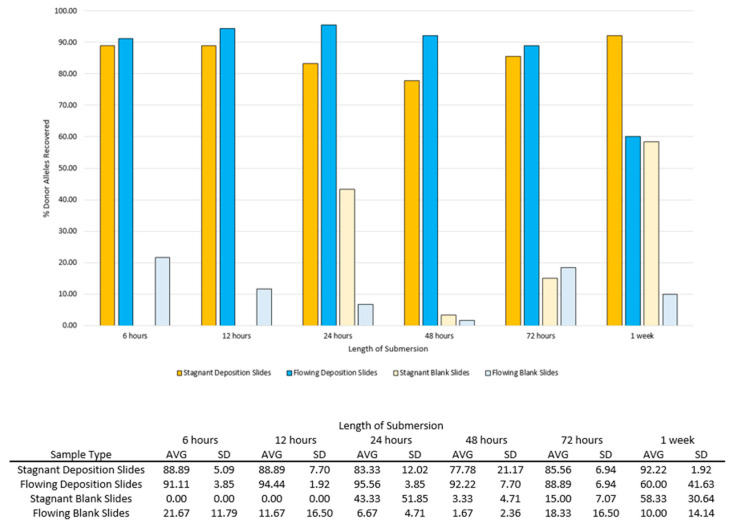
Average (AVG) and standard deviation (SD) of the percent of donor alleles recovered from deposition and blank slide samples in stagnant and flowing conditions associated with Phases 1 and 2.

**Figure 5 genes-14-01045-f005:**
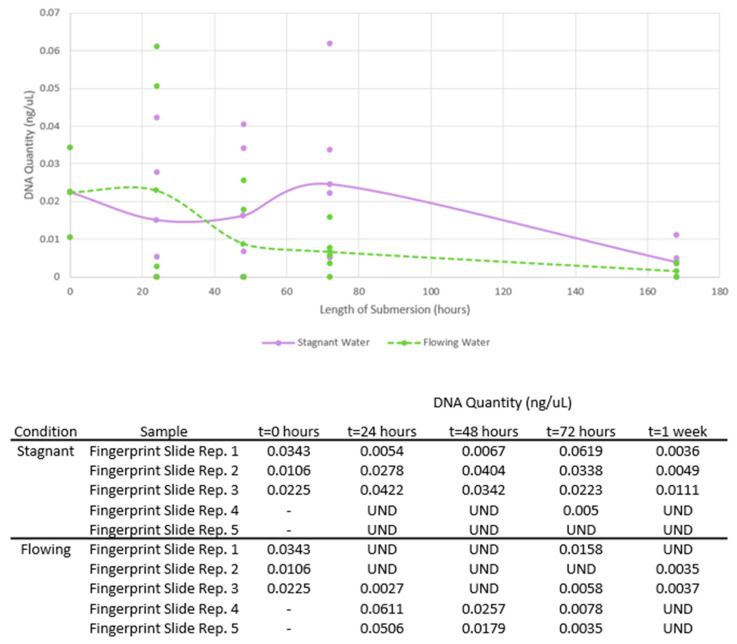
DNA quantities of fingerprint slide samples in stagnant and flowing water over one week submersion. Dots represent individual quantity values for all samples, while lines connect the average quantity values per submersion time. The table shows the DNA quantity obtained for each replicate sample. “UND” refers to an undetected value reported at quantification.

**Figure 6 genes-14-01045-f006:**
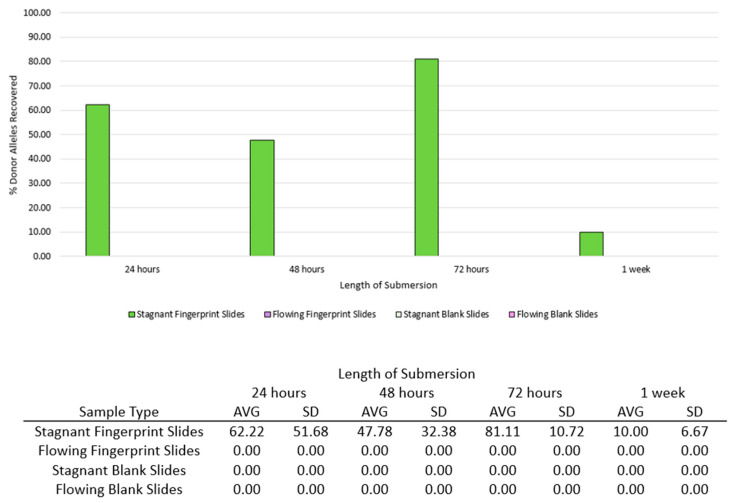
Average (AVG) and standard deviation (SD) of percent of donor alleles recovered from fingerprint and blank slide samples in stagnant and flowing conditions associated with Phases 3 and 4.

**Figure 7 genes-14-01045-f007:**
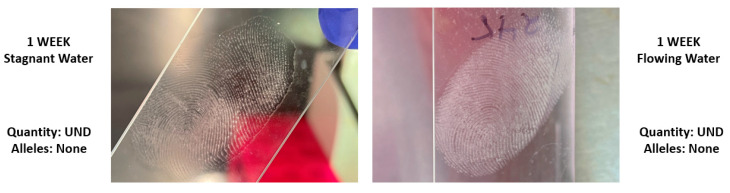
Examples of fingerprint slide samples recovered from stagnant (**left**) and flowing (**right**) water without enhancement. Neither of the pictured fingerprints yielded DNA quantities or amplification products.

**Figure 8 genes-14-01045-f008:**
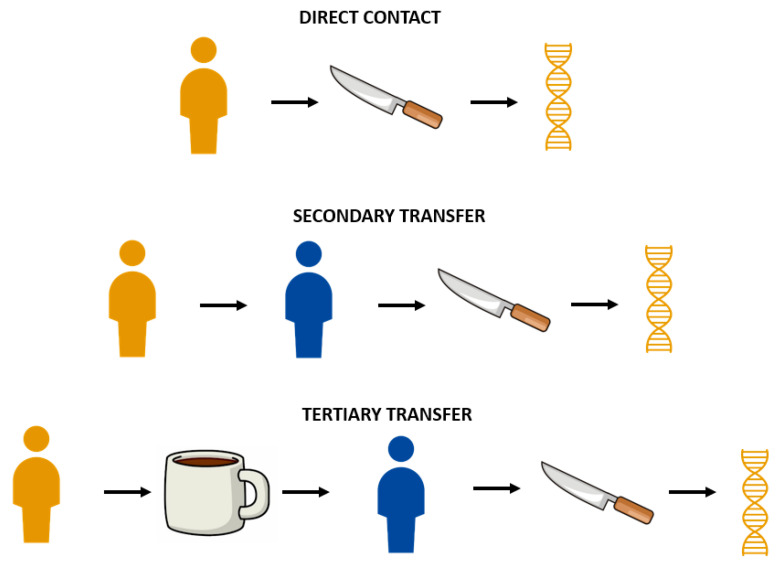
Visual depiction of direct contact (also referred to as primary transfer), secondary transfer, and tertiary transfer of DNA.

**Figure 9 genes-14-01045-f009:**
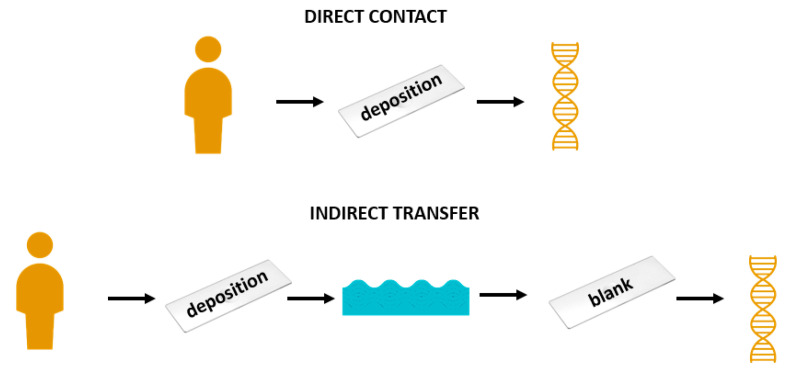
Visual depiction of direct contact and indirect transfer as related to deposition and blank slides associated with Phases 1 and 2.

**Figure 10 genes-14-01045-f010:**
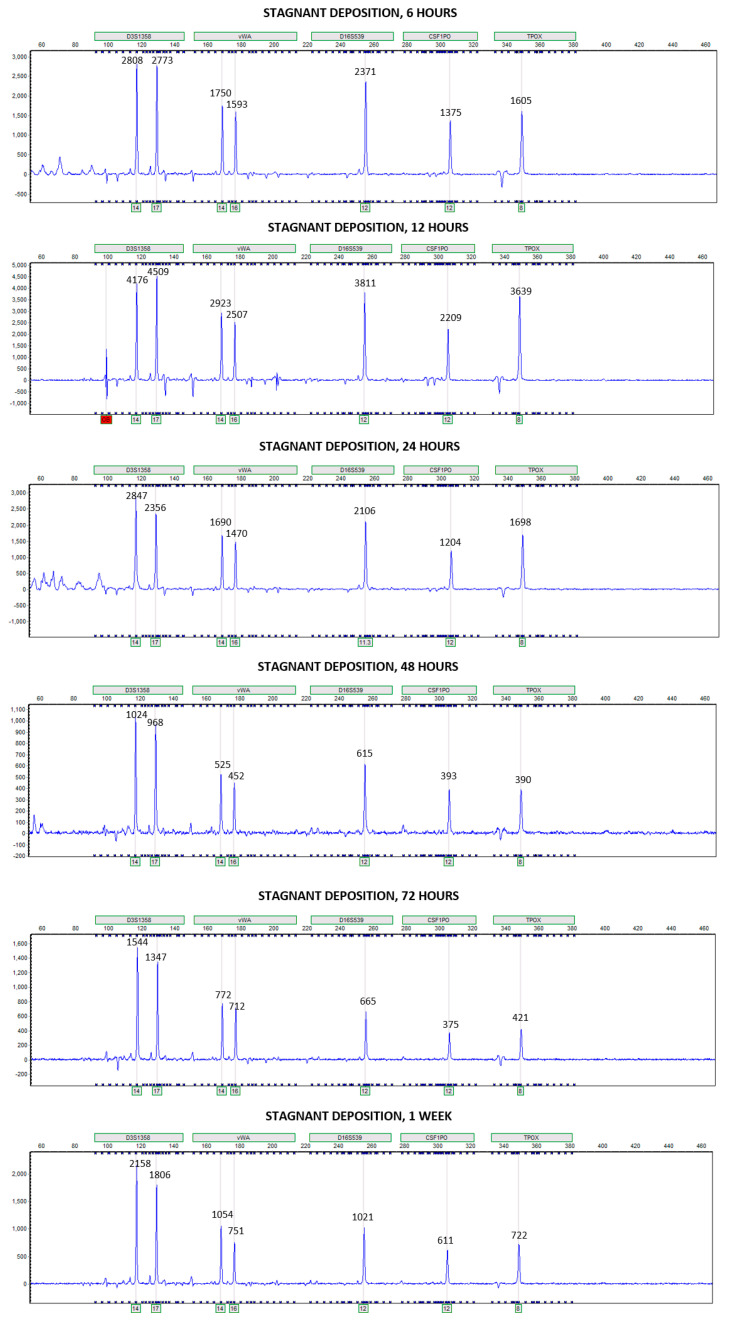
Examples of electropherograms generated at each submersion time for deposition slide samples in stagnant water for Phase 1. Individual peak heights (in RFUs, Relative Fluorescence Units) are labeled for the donor alleles at loci D3S1358, vWA, D16S539, CSF1PO, and TPOX.

**Figure 11 genes-14-01045-f011:**
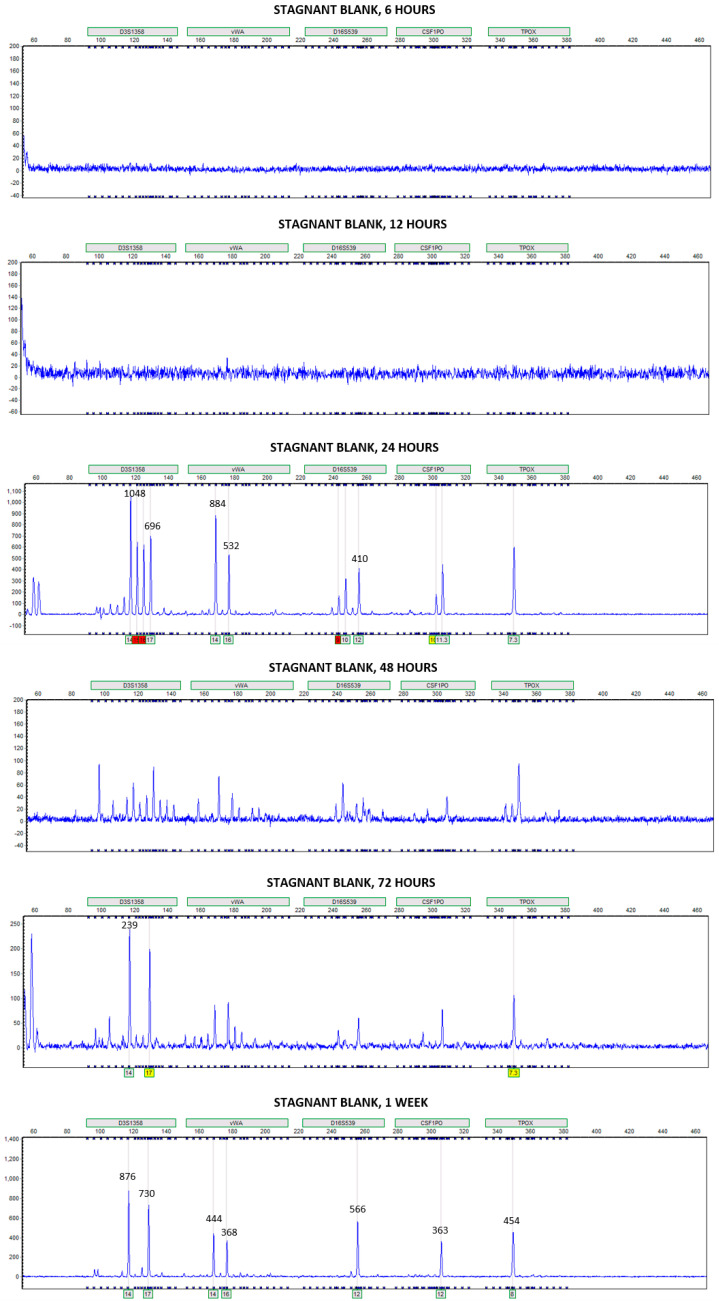
Examples of electropherograms generated at each submersion time for blank slide samples in stagnant water for Phase 1. Individual peak heights (in RFUs, Relative Fluorescence Units) are labeled for the donor alleles at loci D3S1358, vWA, D16S539, CSF1PO, and TPOX.

## Data Availability

The data presented in this study are available on request from the corresponding author. The data are not publicly available due to privacy concerns.

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
