# Peer review of "Preliminary Study: DNA Transfer and Persistence on Non-Porous Surfaces Submerged in Spring Water"

_genes, 2023, doi:10.3390/genes14051045_

Round 1

Reviewer 1 Report

In the manuscript entitled “DNA Transfer and persistence on non-porous surfaces submerged in spring water”, the authors investigate that the deposition of DNA onto glass and ulterior recovery and amplification after submersion in water. Although the topic addressed is interesting, there are some relevant issues that need to be properly assessed:

Major comments

1.      Number of replications should be increased in the case of blank samples.

2.      Data o DNA recovery should be presented row as well as median and interquartile range.

3.      A paragraph explaining the statistical methodology should appear in the methods section.

4.      It is not clear whether DNA quantity recovered, especially in the blank samples is above the limit of detection of the technique. The threshold for the quantification kit is has been indicated to be 16 pg/ul, however, row data about the DNA amount is not shown. It is not clear that in Figure 2, DNA samples are above this limit.

5.      Since nothing is indicated about the statistical method, it results surprisingly that differences in one week between stagnant and flowing water for blank samples do not result statistically significant. This reinforces the idea that detected DNA perhaps is not relevant in any blank sample.

6.      It is mentioned that any foreign alleles detected were not included in calculations. How the detection of “donor alleles” can be unequivocally due to donor and not deposited randomly as the foreign alleles do?

7.      In table 1 all data should be included or, at least, mean and interquartile range.

8.      Figures 7 and 8 do not add any value and should be removed.

9.      Figure 6 legend indicates that “neither fingerprint yelded DNA quantities or amplification products”. This is contradictory with sections 3.3 and 3.4. Please explain.

10.  Discussion section is too large and somehow speculative.

11.  As authors refer in the limitations section, more replicates should be included in order to achieve robust statistical data. Different biological fluids as blood or semen can also be tested. In addition, the extrapolation of these data in the “natural environment” is difficult, since the participation of foreign DNA from living organisms as well as different effects in the deposited samples cannot be discarded.

Author Response

Thank you for your feedback; it is very appreciated.

1.Number of replications should be increased in the case of blank samples.

To this point, we do agree that having a greater number of replicates would have been helpful in this study. Originally, we had planned for the blank samples to serve as controls and were not anticipating to observe an increase in quantity and amplification product on the blank samples. We have decided to remove any statistics for the blank samples and have provided this information simply as observation instead. We also have added "preliminary study" to the title due to the limited number of replicates.

2.Data o DNA recovery should be presented row as well as median and interquartile range.

To this point, we have adjusted the figures for DNA recovery to include both graphs and tables to list out the individual datapoints. Additionally, mean and standard deviation/variance values are present in the manuscript in the paragraphs referring to the figures.

3.A paragraph explaining the statistical methodology should appear in the methods section.

To this point, we agree and have added this into the methods section.

4.It is not clear whether DNA quantity recovered, especially in the blank samples is above the limit of detection of the technique. The threshold for the quantification kit is has been indicated to be 16 pg/ul, however, row data about the DNA amount is not shown. It is not clear that in Figure 2, DNA samples are above this limit.

As we mentioned with our response to #2, we have altered the figure (now Figure 3) to incorporate a table in addition to the graph to make the reported DNA quantities clearer.

5.Since nothing is indicated about the statistical method, it results surprisingly that differences in one week between stagnant and flowing water for blank samples do not result statistically significant. This reinforces the idea that detected DNA perhaps is not relevant in any blank sample.

To this point, any information relating to this has been removed since we are no longer including statistical analyses about the blank samples. 

6. It is mentioned that any foreign alleles detected were not included in calculations. How the detection of “donor alleles” can be unequivocally due to donor and not deposited randomly as the foreign alleles do?

To this point, we do agree that the detection of some "donor alleles" may have been randomly deposited alleles that the donor shared and not unequivocally transferred from the donor. We have expanded on the mention of foreign alleles to clarify that when alleles inconsistent with the donor, deemed "foreign alleles" were detected, they were always accompanied by alleles consistent with the donor. Additionally, the signal produced from these foreign alleles was always lower than that of the alleles consistent with the donor, so for the purposes of counting if a donor allele was present or not, we assumed that foreign and donor alleles were not shared.

7.In table 1 all data should be included or, at least, mean and interquartile range.

To this point, Table 1 has actually been removed in editing.

8.Figures 7 and 8 do not add any value and should be removed.

To this point, we would like to advocate for keeping these figures (now Figures 8 and 9). Although DNA transfer is becoming more widely talked about in the field of forensic science, it is our experience that many people still struggle to understand the concept. We do believe that these figures may add value for those not as familiar with DNA transfer.

9.Figure 6 legend indicates that “neither fingerprint yelded DNA quantities or amplification products”. This is contradictory with sections 3.3 and 3.4. Please explain.

To this point, we have clarified that neither fingerprint in the figure yielded DNA quantities or amplification products. Because we have added in tables listing out the DNA quantities of individual datapoints for sections 3.3 and 3.4, we hope that this clarifies the figure.

10.Discussion section is too large and somehow speculative.

To this point, we were able to trim down the discussion section a bit. We would also like to advocate that some of the discussion around our results, particularly the proposed mechanism of DNA transfer, is inherently speculative due to the unpredictability associated with DNA transfer. However, we do hope that including "preliminary study" in the title supports some of the speculation behind the discussion section.

11.As authors refer in the limitations section, more replicates should be included in order to achieve robust statistical data. Different biological fluids as blood or semen can also be tested. In addition, the extrapolation of these data in the “natural environment” is difficult, since the participation of foreign DNA from living organisms as well as different effects in the deposited samples cannot be discarded.

To your points for this section, we agree about being to further expand on this research, and we hope that future studies are able to do so. Although submerged evidence in a natural environment is subject to a variety of factors, including what you mentioned, we hope that this study can at least start conversations about the ability of DNA to both transfer and persist on these surfaces. Because each natural environment is unique, we at least hope the effects of factors like flow rate, fresh versus salt water, depth, etc. on trace DNA could be used as a guide for investigators collecting this type of evidence.

Once again, we would like to thank you for your helpful feedback, and we hope that this has addressed all your points.

Reviewer 2 Report

The paper deals with an interesting topic. I think that the results her presented may be useful and therefore the perspective provided by the article is certainly of interest in the field of forensic science and in particular in forensic genetics. The manuscript is well organized, the writing is clear and concise, the methology is robust and well described, the results described are very interesting and deserve future developments.

I only think that the bibliography should be updated a little, only in 2022 a lot of studies on touch DNA were published, here are some examples

1. Touch DNA Sampling Methods: Efficacy Evaluation and Systematic Review.

Tozzo P, Mazzobel E, Marcante B, Delicati A, Caenazzo L.Int J Mol Sci. 2022 Dec 8;23(24):15541. doi: 10.3390/ijms232415541.

2. Individual shedder status and the origin of touch DNA.

Jansson L, Swensson M, Gifvars E, Hedell R, Forsberg C, Ansell R, Hedman J.Forensic Sci Int Genet. 2022 Jan;56:102626. doi: 10.1016/j.fsigen.2021.102626.

3. A review on touch DNA collection, extraction, amplification, analysis and determination of phenotype. Nimbkar PH, D Bhatt V.Forensic Sci Int. 2022 Jul;336:111352. doi: 10.1016/j.forsciint.2022.111352. 

4. Detection of touch DNA evidence on swab by SYBR®Green I Nucleic Acid Gel Stain.

Panjaruang P, Romgaew T, Aobaom S.Forensic Sci Int. 2022 Dec;341:111477. doi: 10.1016/j.forsciint.2022.111477.

Author Response

Thank you very much for your feedback; we truly appreciate it. 

To your point of the bibliography needing updating, we did agree. We actually found all four of the articles you recommended very helpful and were able to incorporate them, which added a few more references from 2022. 

Reviewer 3 Report

Dear Authors,

the study is innovative and interesting ,but in my opinion the manuscript should be improved, according to the following suggestions:

1) Par 2.3. Creating and Depositing Samples.

- Clarify if saliva and fingerprint s donor was male or female and the age.

sample were collected according to an informed consent

- Indicate the neat saliva amount extracted and quantified before deposition, and the DNA quantity recovered.

- Indicate the DNA quantity recovered  from 3 fingerprints prior to submersion experiment and  why were not analyzed all 5 fingerprints.  Clarify if all prints were from all from right hand

- Indicate the Glass microscope slides Producer

 2) Par. 2.4. Sample Submersion and Recovery

- Indicate water temperature and indoor environmental temperature at which the experiments have been carried out.

- An image showing  the two different sample  submersion conditions should be added as representative.

3) Par. 2.5. DNA Analysis

 - Describe the procedure used for DNA extraction by Qiagen’s QIAamp DNA Investigator Kit was utilized to extract the DNA collected 147 either from control than from tested samples.

- Internal controls have been used during all steps (extraction, quantification, PCR) in order to monitoring any contamination or error. Please clarify if you have done it.

4) In Par. 3  Results indicate the software or calculation sheet used for the statistical evaluations reported in sub-paragraphs 3.1, 3,2, 3.3, 3.4).

5) Par 5. Conclusions

Due to the Extension of the manuscript, conclusions should be extended.

 6) In Figure 3.  Replace Stagnant Deposition and Flowing Deposition with other terms (i.e. Stagnant condition samples, Flowing Condition samples)  because they may generate confusion. Add the word “sample” to  Stagnant Blank and Flowing Blank.

7) In Table 1 add %

8) The  research is innovative and interesting but in my opinion the study has many limitations due  to the limited number of samples analyzed, the type of surface used,  as reported by the same authors in Par. 4.2.

Because of this I suggest to modify the title including something like “preliminary study” or “proof of concept”.

Finally I suggest to extend the research should be extended including not only more samples and different surfaces but also evaluating the impact produced  by adding soap or detergents to the water or varying water temperature.

Author Response

Thank you very much for your feedback; we truly appreciate it.

1) Par 2.3. Creating and Depositing Samples.

To your points for this section, we added in details about the donor's gender and age and clarified that samples were collected according to informed consent. We also included the producer of the glass microscope slides and indicated the DNA quantity recovered from the initial set of fingerprints prior to submersion. 

2) Par. 2.4. Sample Submersion and Recovery

To your points for this section, we included the indoor environmental temperature for which the experiments were carried out. The actual water temperature was not monitored, so we were unable to add that in. We also entirely agreed that an image of the submersion environments was necessary, and that has been added in as the new Figure 1.

3) Par. 2.5. DNA Analysis

To your points in this section, we described the QIAamp DNA Investigator Kit procedure used and stated which QIAamp protocol was followed. We also clarified that internal controls were used in all steps to monitor contamination and error.

4) In Par. 3  Results indicate the software or calculation sheet used for the statistical evaluations reported in sub-paragraphs 3.1, 3,2, 3.3, 3.4).

To this point, we have added a section into the methodology to indicate the statistical methods utilized for all results.

5) Par 5. Conclusions

We agreed with your point to extend the conclusion section and have fleshed this out more while also hopefully making the conclusions clearer.

6) In Figure 3.  Replace Stagnant Deposition and Flowing Deposition with other terms (i.e. Stagnant condition samples, Flowing Condition samples)  because they may generate confusion. Add the word “sample” to  Stagnant Blank and Flowing Blank.

To your points about Figure 3, the image has been updated (now Figure 4). We have broken the terms up into Stagnant Deposition Slides, Stagnant Blank Slides, Flowing Deposition Slides, and Flowing Blank Slides, as those are the terms used throughout the manuscript. We hope this makes the figure clearer.

7) In Table 1 add %

When the manuscript was updated, Table 1 was removed.

8) The  research is innovative and interesting but in my opinion the study has many limitations due  to the limited number of samples analyzed, the type of surface used,  as reported by the same authors in Par. 4.2. Because of this I suggest to modify the title including something like “preliminary study” or “proof of concept”.

To your points here, we do agree that the study has limitations in the number of replicates per time period, as we also mentioned in the manuscript. I do want to highlight that we did have 140 total data points for quantification and generated electropherograms for 100 samples. However, we added in "preliminary study" as part of the title due to the limited number of replicates.

To your final point about incorporating soap or detergents and water temperature as additional variables, we found this comment very interesting! As mentioned in the manuscript, submerged evidence can be found at indoor or outdoor crime scenes. Much of the future studies section mentioned environmental variables that are more relevant to outdoor crime scenes, yet exploring the effect of detergents mixed with water is also relevant for those indoor scenes. We are aware that research has shown the ability of sperm cells to persist after laundering, but we would be very interested to see how DNA from other cell types may persist, especially if introduced to detergent without as much agitation as the laundering process induces.

Once again, thank you for your helpful feedback, and we believe that everything has been addressed.